# Health-related quality of life, functional impairment and comorbidity in people with mild-to-moderate chronic kidney disease: a cross-sectional study

Simon DS Fraser [ID],[1] Jenny Barker,[1] Paul J Roderick,[1] Ho Ming Yuen,[1] Adam Shardlow,[2] James E Morris,[1] Natasha J McIntyre,[2] Richard J Fluck,[2] Chris W McIntyre,[3] Maarten W Taal [ID] [2,4]

[1]School of Primary Care, Population Sciences and Medical Education, Faculty of Medicine, University of Southampton, Southampton, UK
[2]The Department of Renal Medicine, Royal Derby Hospital NHS Foundation Trust, Derby, UK
[3]Department of Medical Biophysics, University of Western Ontario, London, Ontario, Canada
[4]Division of Medical Sciences and Graduate-Entry Medicine, University of Nottingham, Derby, UK

**Correspondence to**
Dr Simon DS Fraser;
S.Fraser@soton.ac.uk

## ABSTRACT

**Objectives** To determine the associations between comorbidities, health-related quality of life (HRQoL) and functional impairment in people with mild-to-moderate chronic kidney disease (CKD) in primary care.

**Design** Cross-sectional analysis at 5-year follow-up in a prospective cohort study.

**Setting** Thirty-two general practitioner surgeries in England.

**Participants** 1008 participants with CKD stage 3 (of 1741 people recruited at baseline in the Renal Risk in Derby study) who survived to 5 years and had complete follow-up data for HRQoL and functional status (FS).

**Primary and secondary outcome measures** HRQoL assessed using the 5-level EQ-5D version (EQ-5D-5L, with domains of mobility, self-care, usual activities, pain/discomfort and anxiety/depression and index value using utility scores calculated from the English general population), and FS using the Karnofsky Performance Status scale (functional impairment defined as Karnofksy score ≤70). Comorbidity was defined by self-reported or doctor-diagnosed condition, disease-specific medication or blood result.

**Results** Mean age was 75.8 years. The numbers reporting some problems in EQ-5D-5L domains were: 582 (57.7%) for mobility, 166 (16.5%) for self-care, 466 (46.2%) for usual activities, 712 (70.6%) for pain/discomfort and 319 (31.6%) for anxiety/depression. Only 191 (18.9%) reported no problems in any domain. HRQoL index values showed greater variation among those with lower FS (eg, for those with Karnofsky score of 60, the median (IQR) EQ-5D index value was 0.45 (0.24 to 0.68) compared with 0.94 (0.86 to 1) for those with Karnofsky score of 90). Overall, 234 (23.2%) had functional impairment.

In multivariable logistic regression models, functional impairment was independently associated with experiencing problems for all EQ-5D-5L domains (mobility: OR 16.87 (95% CI 8.70 to 32.79, p<0.001, self-care: OR 13.08 (95% CI 8.46 to 20.22), p<0.001, usual activities: OR 8.27 (95% CI 5.43 to 12.58), p<0.001, pain/discomfort: OR 2.94 (95% CI 1.86 to 4.67), p<0.001, anxiety/depression: 3.08 (95% CI 2.23 to 4.27), p<0.001). Higher comorbidity count and obesity were independently associated with problems in mobility, self-care, usual activities and pain/discomfort: for three or

## Strengths and limitations of this study

► This study involved a large cohort of people with chronic kidney disease (CKD) recruited from primary care, a setting in which patients with mild-to-moderate CKD are typically managed in the UK.

► A broad range of comorbidities were included but they were identified at baseline only, not at follow-up, by which time the number of comorbidities may have changed.

► Health-related quality of life and functional status were measured in the same patient group and the use of the EQ-5D-5L index measure and data from the Health Survey for England enabled comparison with a general population.

► Health-related quality of life and functional status measures were taken at 5-year follow-up but not at baseline and we were therefore unable to identify change over time.

► This was a cross-sectional study of survivors and we are therefore not able to draw causative links.

more comorbidities versus none: (mobility: OR 2.10 (95% CI 1.08 to 4.10, p for trend 0.002, self-care: OR 2.64 (95% CI 0.72 to 9.67, p for trend 0.05), usual activities: OR 4.20 (95% CI 2.02 to 8.74, p for trend <0.001), pain/discomfort: OR 3.06 (95% CI 1.63 to 5.73, p for trend <0.001)), and for obese (body mass index (BMI) ≥30 kg/m²) versus BMI <25 kg/m²: (mobility: OR 2.44 (95% CI 1.61 to 3.69, p for trend <0.001), self-care: OR 1.98 (95% CI 1.06 to 3.71, p for trend 0.003), usual activities: OR 1.82 (95% CI 1.19 to 2.76, p for trend 0.019), pain/discomfort: OR 2.37 (95% CI 1.58 to 3.55, p for trend <0.001)). Female sex, lower FS and lower educational attainment were independently associated with anxiety/depression (ORs 1.60 (95% CI 1.18 to 2.16, p 0.002), 3.08 (95% CI 2.23 to 4.27, p<0.001) and 1.67 (95% CI 1.10 to 2.52, p 0.009), respectively). Older age, higher comorbidity count, albuminuria (≥30 mg/mmol vs <3 mg/mmol), lower educational attainment (no formal qualifications vs degree level) and obesity were independently associated with functional impairment (ORs 1.07 (95% CI 1.04 to 1.09, p<0.001), 2.18 (95% CI 0.80 to 5.96, p for trend <0.001),

BMJ

1.74 (95% CI 0.82 to 3.68, p for trend 0.005), 2.08 (95% CI 1.26 to 3.41, p for trend <0.001) and 4.23 (95% CI 2.48 to 7.20), respectively).
**Conclusions** The majority of persons with mild-to-moderate CKD reported reductions in at least one HRQoL domain, which were independently associated with comorbidities, obesity and functional impairment.
**Trial registration number** National Institute for Health Research Clinical Research Portfolio Study Number 6632.

## INTRODUCTION

Chronic kidney disease (CKD) is common globally, affecting about 13% of the general adult population, with CKD stage 3 the most prevalent category.[1 2] Current treatment guidelines for CKD are disease specific and focus on reducing progression and preventing complications such as cardiovascular disease.[3] However, in the UK, most people with CKD stage 3 are managed in primary care and in this context only a minority (18%) evidence progression over 5 years.[4] The risk of end-stage kidney disease (ESKD) is extremely low (0.2%).[4]

Conversely, comorbidities (additional chronic diseases) are common in individuals with CKD and can worsen clinical outcomes and health-related quality of life (HRQoL).[5] Ninety-six per cent of people with stage 3 disease have at least one comorbidity, around 40% have a comorbidity count of two or more.[6]

A significant body of research has explored HRQoL and the functional status (FS) of people with ESKD or following kidney transplant but these factors are not well explored in those with less severe CKD. Among 733 people with high-risk CKD in the Renal Impairment In Secondary Care study, 555 (76%) reported problems in one or more of the EQ5D domains.[7 8]

This is a clinically important knowledge gap because mild-to-moderate reductions in glomerular filtration rate (GFR) are usually asymptomatic, so improved understanding of the comorbidities and symptoms that affect HRQoL and FS in this group of people is important to facilitate a holistic approach to management. The objective of this study was therefore to evaluate HRQoL and determine the associations between comorbidities, HRQoL and functional impairment in people with mild-to-moderate CKD in primary care.

## MATERIALS AND METHODS

A detailed description of the Renal Risk in Derby (RRID) study methodology has been published elsewhere.[9] In summary, approximately 8280 people with CKD stage 3 were identified from renal registers at 32 primary care clinics in Derbyshire, UK, between 2008 and 2010 and invited to participate in the study. Of these people, 1822 attended initial baseline visits and 1741 met eligibility criteria (age ≥18 years; two estimated GFR (eGFR) results (derived from the Modification of Diet in Renal Disease study (MDRD) equation) of 30–59 mL/min/1.73 m$^2$, at least 90 days apart).[9] People with a life expectancy of less than 1 year, who were unable to attend study visits, or who had a solid organ transplant were excluded.

### Health-related quality of life

HRQoL was assessed at 5-year follow-up using the EQ-5D-5L, a widely used, validated measure of health status that can be standardised to different populations. EQ-5D-5L consists of two aspects: a descriptive system, in which participants are asked to rate their health state from 1 to 5 against five domains (mobility, self-care, usual activities, pain/discomfort, anxiety/depression) and the EQ visual analogue scale (EQ-VAS), in which participants rate their health on a scale ranging from 'the best health you can imagine' (100) to 'the worst health you can imagine' (0).[10] An EQ-5D-5L value set has previously been published for England.[11] However, concerns have been raised about the quality and reliability of the data collected in the valuation study such that the English National Institute for Health and Care Excellence (NICE) recommend 'If data were gathered using the EQ-5D-5L descriptive system, utility values in reference-case analyses should be calculated by mapping the 5L descriptive system data onto the 3L value set'.[12] For these analyses, individual health states were therefore converted using the EuroQol EQ-5D-5L Crosswalk Index Value Calculator into a single 3L index value (a preference-based score that typically ranges from states worse than dead (<0) to 1 (full health) with dead at 0) using utility scores calculated from the English general population.[13 14] The index value and the EQ-VAS score were used to graphically display the relationship between HRQoL and FS.

### Functional status

FS, defined in this paper as the physical ability to perform normal activities and independently self-care, was assessed at 5-year follow-up using the Karnofsky Performance Status (KPS) scale. The KPS is a clinician-assessed score originally developed in oncology and was used for assessing prognosis and management in patients with cancer.[15] The scale ranges from 'normal/no complaints' (100) to 'dead' (0). Theoretically, the scale can take any whole number value within the range, but in practice results are commonly recorded as multiples of 10; therefore, KPS was treated as an ordinal variable in this study. The original continuous KPS score >70 is defined as being 'able to carry on normal activity and to work with no special care needed', a score of between 50 and 70 inclusive is defined as 'unable to work, able to live at home and care for most personal needs; varying amount of assistance needed', and a score of less than or equal to 40 is defined as 'unable to care for self, requiring the equivalent of institutional or hospital care'.[15] Functional impairment was analysed as a binary outcome due to the small number of patients with low KPS score. A KPS score of ≤70 versus >70 was chosen to compare those able to carry on normal life with those experiencing some functional impairment as has been used in evaluation of FS in patients with lung cancer.[16]

## Comorbidities identified at baseline

The methods for defining comorbidities in participants have been described in detail elsewhere.[6] In brief, eleven comorbidities were pragmatically identified at baseline using information from a combination of sources and agreed by consensus between three clinicians (SF, MWT and PJR): patient questionnaires in which patients were asked to list chronic medications (followed by verbal confirmation with verification of repeat prescriptions where possible), blood pressure measurement at the time of baseline study visit and self-reported clinical diagnoses. Self-reported comorbidities included heart failure, ischaemic heart disease, peripheral vascular disease (defined as peripheral arterial revascularisation or amputation) and cerebrovascular disease (stroke or transient ischaemic attack). Diagnoses of chronic respiratory disorder, depression, painful condition, hypertension, diabetes and thyroid disorders were made according to medication history or patient report. Anaemia was defined according to Kidney Disease Improving Global Outcomes (KDIGO) guidelines as haemoglobin <13.0 g/dL (<130 g/L) in men and <12.0 g/dL (<120 g/L) in women, at baseline.[17] Hypertension was defined either by medication history, or by a systolic blood pressure >140 mm Hg or diastolic >90 mm Hg, at baseline.

## Kidney function

Kidney function was assessed at 5-year follow-up. eGFR was calculated using the Chronic Kidney Disease Epidemiology Collaboration equation and was treated as a continuous variable. The urine albumin-to-creatinine ratio (uACR) from three consecutive early morning specimens was used for analysis. uACR was categorised into three levels according to KDIGO guidelines and fitted as a discrete variable in regression analyses.

Methods for defining CKD progression have also been detailed elsewhere.[4] In summary, progression of CKD was defined as a 25% decline in GFR, coupled with a worsening of GFR category or an increase in albuminuria category. CKD remission was defined as the presence of both eGFR >60 mL/min/1.73 m$^2$ and uACR <3 mg/mmol at any study visit in an individual who had previously met KDIGO diagnostic criteria for CKD.

## Other baseline measures

Body mass index (BMI) was calculated from weight in kilograms divided by square of height in metres and was treated as a categorical variable.[18] Smoking status was categorised as never smoked, ex-smoker and current smoker. Socioeconomic status was assessed using self-reported educational attainment (categorised into no formal qualifications, school or equivalent qualifications, and degree or equivalent qualifications) as well as the Index of Multiple Deprivation (IMD) score, categorised in quintiles.[19] The IMD is a measure of relative deprivation for small areas of residence in England and combines information from seven domains: income; employment; education, skills and training; health and disability; crime; barriers to housing and services; and living environment. Self-reported ethnicity status was also collected.

## Statistical analyses

Descriptive statistics were used to show the characteristics of the study participants at 5-year follow-up. Descriptive statistics were also used to show the distribution of functional impairment (KPS ≤70) among those reporting problems in the five EQ-5D-5L domains. Associations between the patient-reported EQ-5D-5L domains and FS was assessed using the $\chi^2$ test. Ratings from the five participant-reported EQ-5D-5L domains were also compared between the RRID cohort and those reported by people aged 65 years and over in the 2012 Health Survey for England (HSE)—which is representative of the England population.[20] A comparison of basic characteristics was also made between those with and without complete 5-year follow-up data.

Univariable logistic regression models were used to assess the relationships between having 'some problems' in each EQ-5D-5L domain and each predictor variable, including comorbidity count and year-five eGFR. Variables considered to be clinically relevant and where p<0.1 on univariable analysis were subsequently included in multivariable logistic regression models. This process was then repeated for the relationship with the outcome variable functional impairment. Due to the small number of non-white participants, ethnicity was not included in the models.

In the regression models, interactions between the individual and area measures of socioeconomic status were also tested because of the potential for the relationship between individual socioeconomic status (indicated by educational attainment) and HRQoL to vary by area deprivation, particularly for older people.[21] The level of significance was set at 5%. All analyses were performed using Stata/IC V.15.0.

## Patient and public involvement

The RRID study design was discussed with a patient and two feedback meetings for participants and their families were organised after the year 5 visits, which were well attended. In addition, a web page provides updates and information for participants (https://www.uhdb.nhs.uk/renal-risk-in-derby-rrid-study/).

## RESULTS

Of 1741 participants recruited, 1494 survived to 5 years, and of these 1008 participants (67% of survivors) had complete 5-year follow-up data for HRQoL and FS (figure 1). The mean age of the cohort was 75.8 years (SD 8.6) and the majority (n=621, 61.6%) were female (table 1).

Approximately half (n=506, 50.3%) reported having had no formal education, just under half (n=497, 49.4%) lived in areas of lower deprivation (IMD quintiles four or five) and the majority (n=994, 98.6%) were white. The

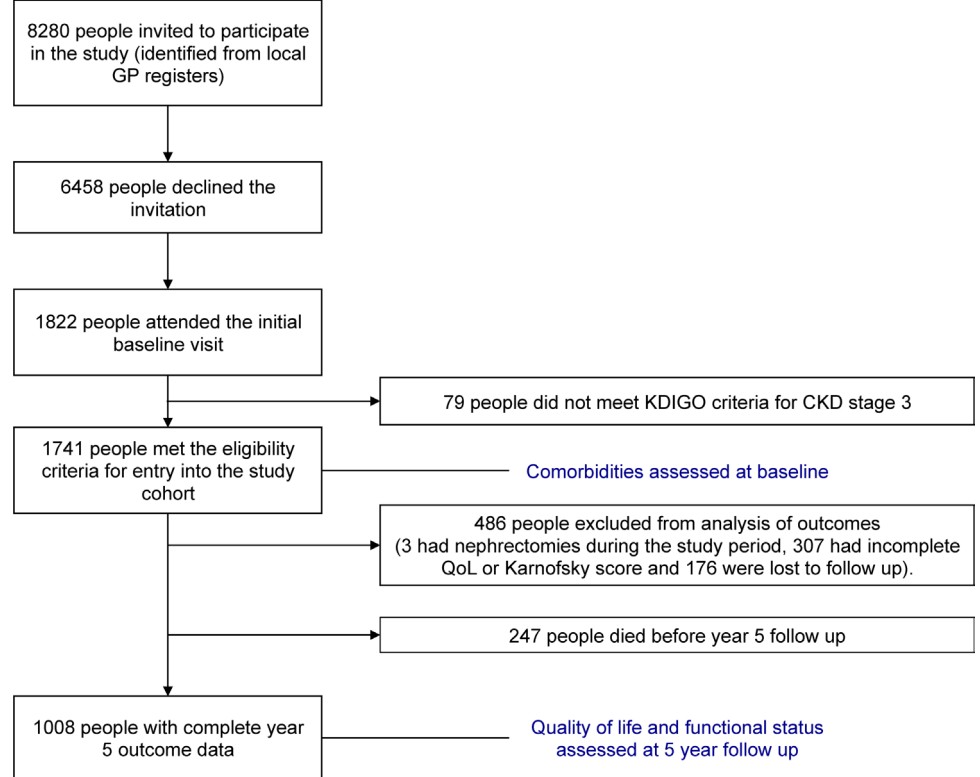

**Figure 1** Flow chart of study participants. CKD, chronic kidney disease; KDIGO, Kidney Disease Improving Global Outcomes; GP, general practitioner; QoL, quality of life.

mean eGFR at follow-up was $54.0\,\mathrm{mL/min/1.73\,m^2}$ (SD 15.2) and almost half (n=460, 45.6%) had had stable CKD over the preceding 5-year period. Only 56 (5.6%) had no comorbidities and about a third (n=344, 34.1%) had three or more comorbidities. For comparison of basic characteristics of those with and without complete 5-year follow-up data, see online supplementary table S1. A slightly higher proportion of those with incomplete follow-up data had three or more comorbidities and only a very small proportion had functional impairment (online supplementary table S1).

The majority reported some impairment in HRQoL overall, with a median score of 75 out of 100 (IQR 60–90) on the EQ-VAS. A minority (n=378, 37.5%) had an EQ-5D-3L index score higher than the age/sex matched median, and only 18.9% of people (n=191) reported no problems across any of the individual HRQoL domains. Furthermore, a majority of participants reported some problems with mobility (n=582, 57.7%) and pain/discomfort (n=712, 70.6%; table 1).

When comparing the self-reported HRQoL domains with HSE data, the proportion of people in the RRID population reporting problems with mobility or pain/discomfort was higher (57.7% vs 50.4%, and 70.6% vs 60.1%, respectively) than in the HSE population (table 2).

For clinician-assessed FS, only two participants had performance status assessed as KPS ≤40 ('unable to care for self, requiring the equivalent of institutional or hospice care') and 232 (23%) were assessed as KPS 50–70

('unable to work; able to live at home and care for most personal needs; varying amount of assistance needed').

The association between clinician-assessed FS and patient-reported HRQoL was complex, either when based on the index score (figure 2A) or the VAS scale (figure 2B). HRQoL was generally higher among those with better FS. However, the spread of HRQoL scores (using either of the HRQoL metrics) was broader among those with lower FS, suggesting a greater degree of variation in HRQoL among those with lower FS than among those with higher FS (figure 2). A higher proportion of people with clinician-assessed functional impairment (KPS ≤70) reported having some degree of problems in each of the five EQ-5D-5L domains than people without functional impairment (online supplementary table S2).

Using the mobility domain as an example (table 3), on univariable analysis older age, greater area deprivation level, higher number of comorbidities, poorer FS, lower eGFR, higher level of albuminuria, lower educational attainment and higher BMI were associated with having some problems.

In the fully adjusted multivariable model, these associations remained for older age, higher number of comorbidities, poorer FS and higher BMI (table 3). A summary of the main independent associations identified in the multivariable logistic regression models for usual activities, self-care, pain/discomfort and anxiety/depression is shown in table 4 and the full analyses in online supplementary tables S3 to S6.

**Table 1** Characteristics of patients at 5-year follow-up in the Renal Risk in Derby study, n=1008 unless otherwise stated

| Variable | Category | Descriptive statistics |
|---|---|---|
| Age in years,* mean (SD) | | 75.8 (8.6) |
| Age group,* n (%) | <70 years | 220 (21.8) |
| | 70–80 years | 467 (46.3) |
| | >80 years | 321 (31.8) |
| Sex,† n (%) | Male | 387 (38.4) |
| | Female | 621 (61.6) |
| Ethnicity,† n (%) | White | 994 (98.6) |
| | Other‡ | 14 (1.4) |
| Educational attainment,† n (%) (n=1007) | No formal qualifications | 506 (50.3) |
| | GCSE, A level, NVQ 1–3 | 291 (28.9) |
| | First or higher degree, NVQ 4–5 | 210 (20.9) |
| Index of Multiple Deprivation (IMD quintile relative to England),† n (%) (n=1006) | Quintile 1 (most deprived) | 82 (8.2) |
| | Quintile 2 | 243 (24.2) |
| | Quintile 3 | 184 (18.3) |
| | Quintile 4 | 258 (25.6) |
| | Quintile 5 (least deprived) | 239 (23.8) |
| Body mass index,† n (%) | Normal or underweight (<25 kg/m$^2$) | 195 (19.3) |
| | Overweight (25–29.99 kg/m$^2$) | 422 (41.9) |
| | Obese (≥30 kg/m$^2$) | 391 (38.8) |
| Smoking status,† n (%) | Never smoked | 496 (49.2) |
| | Ex-smoker | 468 (46.4) |
| | Current smoker | 44 (4.4) |
| eGFR* in mL/min/1.73 m$^2$, mean (SD) (n=1007) | | 54.0 (15.2) |
| uACR* in mg/mmol, median (IQR) (n=1007) | | 0.7 (0–3.3) |
| KDIGO uACR categories* n (%) | A1 | 741 (73.5) |
| | A2 | 217 (21.5) |
| | A3 | 50 (5.0) |
| KDIGO eGFR categories* (eGFR in mL/min/1.73 m$^2$) | G1 (eGFR ≥90) | 13 (1.3) |
| | G2 (eGFR 60–89) | 337 (33.4) |
| | G3a (eGFR 45–59) | 379 (37.6) |
| | G3b (eGFR 30–44) | 223 (22.1) |
| | G4 (eGFR 15–29) | 55 (5.5) |
| | G5 (eGFR <15) | 1 (0.1) |
| Progression of kidney disease,* n (%) | Stable | 460 (45.6) |
| | Progression | 244 (24.2) |
| | Remission | 304 (30.2) |
| Number of comorbidities,† n (%) | None (CKD only) | 56 (5.6) |
| | One | 308 (30.6) |
| | Two | 300 (29.8) |
| | Three or more | 344 (34.1) |
| Individual comorbidities,† n (%) | Hypertension | 874 (86.7) |
| | Painful condition | 300 (29.8) |
| | Anaemia | 201 (19.9) |
| | Ischaemic heart disease | 187 (18.6) |
| | Diabetes | 143 (14.2) |
| | Thyroid disorder | 127 (12.6) |
| | Cerebrovascular disease | 96 (9.5) |
| | Chronic respiratory disorder | 79 (7.8) |
| | Depression | 59 (5.9) |
| | Peripheral vascular disease | 29 (2.9) |
| | Heart failure | 24 (2.4) |
| Quality-of-life domains (any problems reported in each EQ-5D-5L domain, n (%)) | Mobility problems | 582 (57.7) |
| | Self-care problems | 166 (16.5) |
| | Usual activity problems | 466 (46.2) |
| | Pain/discomfort | 712 (70.6) |
| | Anxiety/depression | 319 (31.6) |
| | No problems in any domain | 191 (18.9) |
| Functional status* (KPS score), n (%) | Functional impairment (KPS ≤70) | 234 (23.2) |
| | KPS >70 (able to carry on normal activity and to work; no special care needed) | 774 (76.8) |

Continued

Where variable category percentages sum to less than or more that 100%, this is due to rounding.
*Variables assessed at year 5 follow-up.
†Variables assessed at baseline.
‡Includes mixed, Asian, Cypriot and other.
CKD, chronic kidney disease; eGFR, estimated glomerular filtration rate; GCSE, General Certificate of Secondary Education; KDIGO, Kidney Disease: Improving Global Outcomes; KPS, Karnofsky Performance Status; A level, advanced level; NVQ, National Vocational Qualifications; uACR, urine albumin-to-creatinine ratio.

Factors associated with a lower FS on univariable analysis included older age, lower socioeconomic status (assessed by both IMD score and educational attainment), higher number of comorbidities, obesity, reduced eGFR and greater degree of albuminuria. Other than reduced eGFR, all of these factors remained significant after adjustment (table 5). No interactions were identified in any analyses.

## DISCUSSION

In this cross-sectional study of people with mild-to-moderate CKD who survived to year 5 in a UK primary care cohort, overall patient-reported HRQoL was relatively high

**Table 2** Comparison of the EQ-5D-5L quality-of-life domains between the Health Survey for England (HSE) 2012 and the Renal Risk in Derby (RRID) cohort

| | HSE 2012 cohort* (n=258) | | RRID CKD cohort (n=1008) | |
|---|---|---|---|---|
| | n | % | n | % |
| **Mobility** | | | | |
| 1 (no problems in walking about) | 128 | 49.6 | 426 | 42.3 |
| 2–5 (some problems) | 130 | 50.4 | 582 | 57.7 |
| **Self-care** | | | | |
| 1 (no problems washing or dressing) | 209 | 81.0 | 842 | 83.5 |
| 2–5 (some problems) | 49 | 19.0 | 166 | 16.5 |
| **Usual activities** | | | | |
| 1 (no problems doing usual activities) | 142 | 55.0 | 542 | 53.8 |
| 2–5 (some problems) | 116 | 45.0 | 466 | 46.2 |
| **Pain/discomfort** | | | | |
| 1 (no pain or discomfort) | 103 | 39.9 | 296 | 29.4 |
| 2–5 (some pain or discomfort) | 155 | 60.1 | 712 | 70.6 |
| **Anxiety/depression** | | | | |
| 1 (not anxious or depressed) | 188 | 72.9 | 689 | 68.4 |
| 2–5 (some anxiety or depression) | 70 | 27.1 | 319 | 31.6 |

*All participants were aged 65 years or above.
CKD, chronic kidney disease.

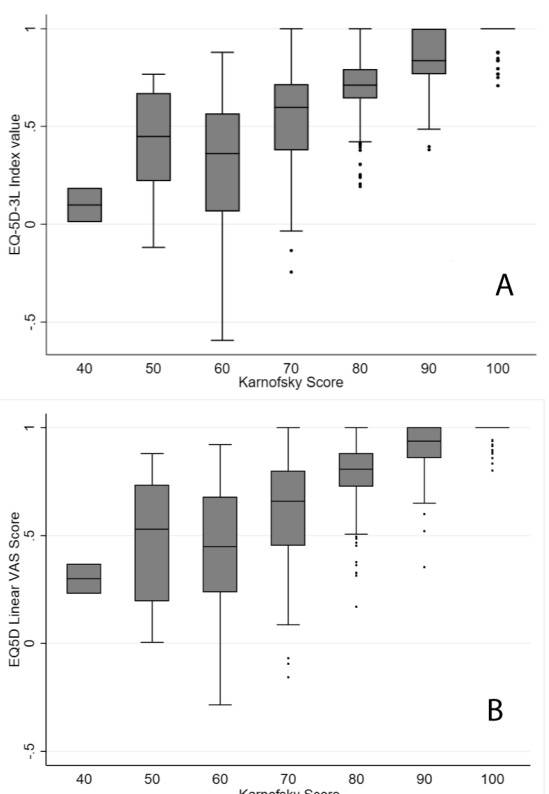

**Figure 2** Relationship between quality of life and functional status. (A) Functional status (by Karnofsky score) and quality of life (by EQ-5D-3L Index score). (B) Functional status (by Karnofsky score) and quality of life (by EQ-5D self-reported visual analogue scale (VAS)).

though a substantial proportion of participants reported problems in each HRQoL domain. A majority reported problems with mobility and pain/discomfort. Although most people had a clinician-assessed FS suggesting that they were able to carry on normal activity and to work with no special care needed, about a quarter were assessed as having functional impairment (being unable to work but able to live at home and care for most personal needs with a varying amount of assistance needed). HRQoL was generally higher among those with better FS but there was more variation in HRQoL among those with lower FS, and low FS was independently strongly associated with low HRQoL in regression analyses. Higher number of comorbidities and obesity were independently associated with problems in most EQ-5D-5L domains and with functional impairment. Functional impairment was independently associated with experiencing some problems across all EQ-5D-5L domains.

This study had several strengths, including the large size of the cohort, and recruitment from primary care, a setting in which patients with mild-to-moderate CKD are typically managed. The RRID cohort is pragmatic and likely to represent a population of typical patients with mild-to-moderate CKD in the UK.[22 23] We were able to identify a broad range of comorbidities but they were identified at baseline only. The number of comorbidities may therefore have changed by the time of follow-up assessment, meaning that our comorbidity prevalence data were likely underestimates of the true prevalence in some patients. Similarly, certain other exposures were assessed at baseline and could potentially have changed by the time of follow-up. We recognise these as important limitations but consider that they are unlikely to significantly alter the main findings of our study with regard to HRQoL and FS. A further strength is that we were able to measure HRQoL and FS in the same patient group and the HRQoL and FS data were relatively complete. The use of the EQ-5D-5L index measure and data from the HSE enabled comparison with a general population. However, the index values for HRQoL required conversion to 3L values as reliable 5L index values are not yet available for all standard populations. Evidence from a previous RRID analysis on prior renal function change provided depth to our analyses for this cross-sectional study. However, there were also several important limitations—this was a

**Table 3** Logistic regression models examining associations between lower quality of life (EQ-5D-5L mobility domain categorised as 'no problems' vs 'any problems') and patient characteristics

| | Univariable | | Multivariable* | |
|---|---|---|---|---|
| | OR (95% CI) | P value | OR (95% CI) | P value |
| Age (years) | 1.05 (1.03 to 1.07) | <0.001 | 1.03 (1.02 to 1.05) | 0.001 |
| Female sex (vs male) | 1.16 (0.89 to 1.49) | 0.27 | – | – |
| **Index of Multiple Deprivation (IMD quintile relative to England) (vs quintile 5: least deprived) (n=1006)** | | | | |
| Quintile 1 (most deprived) | 1.61 (0.96 to 2.69) | 0.04† | 0.95 (0.52 to 1.74) | 0.436† |
| Quintile 2 | 1.69 (1.17 to 2.43) | | 1.38 (0.90 to 2.10) | |
| Quintile 3 | 1.10 (0.75 to 1.62) | | 0.94 (0.60 to 1.48) | |
| Quintile 4 | 1.25 (0.88 to 1.78) | | 1.17 (0.78 to 1.76) | |
| **Number of comorbidities (vs no comorbidities)** | | | | |
| One | 1.25 (0.70 to 2.24) | <0.001† | 1.01 (0.53 to 1.93) | 0.002† |
| Two | 2.43 (1.35 to 4.38) | | 1.40 (0.72 to 2.71) | |
| Three or more | 4.50 (2.49 to 8.12) | | 2.10 (1.08 to 4.10) | |
| **Functional status (KPS score) (vs KPS >70)** | | | | |
| Functional impairment (KPS ≤70) | 26.03 (13.60 to 49.81) | <0.001 | 16.87 (8.70 to 32.79) | <0.001 |
| **eGFR (mL/min/1.73 m$^2$) (N=1007)** | | | | |
| | 0.98 (0.98 to 0.99) | <0.001 | 0.99 (0.99 to 1.01) | 0.983 |
| **uACR (KDIGO categories) (vs category A1,<3 mg/mmol) (n=1007)** | | | | |
| A2 (3–29 mg/mmol) | 1.39 (1.02 to 1.91) | 0.074† | 1.01 (0.69 to 1.48) | 0.937† |
| A3 (≥30 mg/mmol) | 1.42 (0.78 to 2.57) | | 0.88 (0.41 to 1.85) | |
| **Educational attainment (vs first or higher degree or NVQ 4–5) (n=1007)** | | | | |
| No formal qualifications | 2.20 (1.59 to 3.05) | <0.001† | 1.38 (0.93 to 2.04) | 0.238† |
| GCSE, A level or NVQ 1–3 | 1.23 (0.86 to 1.76) | | 1.13 (0.75 to 1.70) | |
| **BMI (vs <25 kg/m$^2$)** | | | | |
| Overweight (BMI 25–29.99 kg/m$^2$) | 1.51 (1.07 to 2.13) | <0.001† | 1.37 (0.93 to 2.01) | <0.001† |
| Obese (BMI ≥30 kg/m$^2$) | 3.24 (2.26 to 4.63) | | 2.44 (1.61 to 3.69) | |
| **Smoking status (vs never smoked)** | | | | |
| Current smoker | 1.50 (0.79 to 2.87) | 0.413† | – | – |
| Ex-smoker | 1.10 (0.85 to 1.42) | | – | |

n=1008 in univariable models unless otherwise stated; n=1005 for final multivariable logistic regression models.
*Adjusted for age, deprivation level, number of comorbidities, functional status, estimated glomerular filtration rate (eGFR) at 5-year follow-up, urinary albumin-to-creatinine ratio (uACR) at 5-year follow-up, educational attainment and body mass index (BMI).
†P value for trend.
GCSE, General Certificate of Secondary Education; KDIGO, Kidney Disease Improving Global Outcomes; KPS, Karnofsky Performance Status; A level, advanced level; NVQ, National Vocational Qualifications.;

cross-sectional study of survivors and we were therefore not able to draw causative links. HRQoL and FS measures were taken at 5-year follow-up but not at baseline and we were therefore unable to identify change over time. The RRID cohort predominantly comprises people of white ethnicity, limiting generalisability of our findings. Comparison with HSE data was undertaken only via univariable analyses, such that potential confounding factors may have influenced the differences observed between the two groups. We also did not have sufficient numbers to allow for reliable exploration of associations between specific comorbidities and HRQoL or FS. We also recognise the need for caution in the interpretation of the associations between functional impairment and problems in individual domains due to small numbers in some individual categories (leading to wide CIs). A further limitation is that one inclusion criterion was the ability to attend study visits, which would have resulted in some selection bias by excluding the very frail.

People with CKD are likely to have multiple comorbidities due both to the nature of the disease process and the relationship between CKD and older age. We have identified that comorbidity count was an independent determinant of both HRQoL and FS, highlighting the importance of a holistic approach that includes attention to comorbidities in the management of people with

**Table 4** Summary matrix of the independent associations with 'some problems' in each domain of the EQ-5D-5L (from multivariable logistic regression analyses)

| | Mobility | Self-care | Usual activities | Pain/discomfort | Anxiety/depression |
|---|---|---|---|---|---|
| Increasing age | O | | | | |
| Female sex | | | | | O |
| Greater area deprivation level | | | | | |
| Higher number of comorbidities | O | O | O | O | |
| Functional impairment | O | O | O | O | O |
| Lower eGFR | | | | | |
| Higher level of albuminuria | | | | | |
| Lower educational attainment | | | | | O |
| Higher BMI | O | O | O | O | |
| Smoking | | O | | | |

BMI, body mass index; eGFR, estimated glomerular filtration rate.

mild-to-moderate CKD. As reported previously, 40% of people in this cohort with stage 3 disease had at least two comorbidities.[6] There are clearly shared risk factors for several of the included comorbidities. It is therefore perhaps unsurprising that in a large cohort of over half a million Canadian patients with CKD, comorbidities such as hypertension and diabetes were common (46.6% and 17.8%, respectively). However, a substantial number of patients also had 'discordant' comorbidities such as chronic pulmonary disease (14.0%) and 10.6% of patients had chronic pain.[5] Comorbidities were all associated with an increased risk of hospitalisation.[5] It is striking that the majority of people in our cohort reported problems with mobility and chronic pain/discomfort, and that both were more prevalent than in a nationally representative sample of the English general population of similar age. About 30% of our cohort were taking analgesic medication, but about 71% reported pain or discomfort in the EQ-5D-5L. This likely reflects the association of CKD with comorbidities, since mild-to-moderate reductions in GFR are unlikely to cause poor mobility or pain. Nevertheless, this observation further highlights the need to pay attention to mobility issues and pain management in order to improve quality of life in people with stage 3 CKD.

The prevalence of diabetes in this population of people with CKD stage 3 was 14.2%. This is lower than the prevalence of 20.1% noted in analyses of CKD prevalence in the HSE.[2] It is possible that this relates to this study population comprising survivors at 5 years in a cohort study and those with diabetes may have been more likely to die prior to these analyses than those without. The study population was also predominantly white and those ethnic groups with greater diabetes prevalence were therefore under-represented.

Mental health problems are common with 26% of adults in England reporting a diagnosis at some point in time.[24] In our study, about 6% of people were classified as having a depressive condition, defined pragmatically based on current antidepressant use or patient self-report,

although about 32% reported some anxiety or depression in the EQ-5D-5L, so this was probably an underestimate. In a large meta-analysis, approximately 25% of adults with CKD stage 1–5 had symptoms suggestive of depression.[25] This appears to persist even in milder forms of the disease and a large study from 2012 showed the prevalence of depression in people with CKD whose eGFR was $\geq 60\,\text{mL/min}/1.73\,\text{m}^2$ was 23.6%.[26] These data imply that careful attention to mental health problems, including screening for depression, may also be key interventions to improve HRQoL in people with mild-to-moderate CKD.

Being able to carry out activities of daily living (ADLs) is an important part of living independently. There is a recognised association between having difficulty with ADLs and a lower HRQoL in older adults, though comparison of HRQoL (as assessed by EQ-5D-5L) with FS (as assessed by the KPS) is understudied.[27] The strong association observed in this study between functional impairment and lower HRQoL including mobility, self-care and usual activities, even after adjustment for potential confounding factors, suggests that clinicians identifying functional impairment should consider their patient's HRQoL. Previous research has identified higher prevalence of problems with ADLs among people with CKD (defined as eGFR $<60\,\text{mL/min}/1.73\,\text{m}^2$) than people without, variously reported as between 26% and 55% depending on the nature of the population studied and the measure of ADL used.[28–30] Two large longitudinal studies have shown, in keeping with our findings, a significant reduction in instrumental ADLs and basic ADLs when renal function deteriorates over time.[28 29] It should also be noted that the KPS is a clinician-assessed tool comprising an element of subjectivity and therefore may be less accurate than an in-depth questionnaire looking at both instrumental ADLs and generic ADLs.

In 2016, NICE published their first guideline for the management of individuals with multiple coexisting chronic diseases.[31] These guidelines ask clinicians to consider the overall burdens of disease and treatment,

**Table 5** Logistic regression models of the associations between clinician-assessed functional impairment (Karnofsky Performance Status score ≤70) and patient characteristics

| | Univariable | | Multivariable* | |
| --- | --- | --- | --- | --- |
| | OR (95% CI) | P value | OR (95% CI) | P value |
| Age (years) | 1.07 (1.05 to 1.09) | <0.001 | 1.07 (1.04 to 1.09) | <0.001 |
| Sex (vs male) | 1.15 (0.85 to 1.56) | 0.371 | 1.32 (0.91 to 1.91) | 0.148 |
| Index of Multiple Deprivation (IMD quintile relative to England) (vs quintile 5: least deprived) (n=1006) | | | | |
| Quintile 1 (most deprived) | 2.77 (1.55 to 4.95) | 0.003† | 2.03 (1.06 to 3.87) | 0.045† |
| Quintile 2 | 2.19 (1.39 to 3.44) | | 2.05 (1.24 to 3.40) | |
| Quintile 3 | 1.83 (1.12 to 2.98) | | 2.02 (1.17 to 3.47) | |
| Quintile 4 | 1.64 (1.03 to 2.59) | | 1.65 (1.00 to 2.75) | |
| Number of comorbidities (vs no comorbidities) | | | | |
| One | 1.18 (0.44 to 3.18) | <0.001† | 0.62 (0.22 to 1.77) | <0.001† |
| Two | 3.10 (1.19 to 8.08) | | 1.22 (0.44 to 3.38) | |
| Three or more | 5.97 (2.32 to 15.35) | | 2.18 (0.80 to 5.96) | |
| eGFR (mL/min/1.73 m$^2$) (n=1007) | | | | |
| | 0.97 (0.96 to 0.98) | <0.001 | 0.99 (0.98 to 1.00) | 0.186 |
| uACR (KDIGO categories) (vs category A1, <3 mg/mmol) (n=1007) | | | | |
| A2 (3–29 mg/mmol) | 1.92 (1.37 to 2.70) | 0.002† | 1.92 (1.29 to 2.87) | 0.005† |
| A3 (≥30 mg/mmol) | 1.96 (1.05 to 3.66) | | 1.74 (0.82 to 3.68) | |
| Educational attainment (vs first or higher degree or NVQ 4–5) (n=1007) | | | | |
| No formal qualifications | 2.77 (1.81 to 4.26) | <0.001† | 2.08 (1.26 to 3.41) | <0.001† |
| GCSE, A level or NVQ 1–3 | 1.07 (0.65 to 1.77) | | 0.99 (0.56 to 1.75) | |
| BMI (vs <25 kg/m$^2$) | | | | |
| Overweight (BMI 25–29.99 kg/m$^2$) | 1.54 (0.94 to 2.53) | <0.001† | 1.59 (0.93 to 2.73) | <0.001† |
| Obese (BMI ≥30 kg/m$^2$) | 3.76 (2.34 to 6.04) | | 4.23 (2.48 to 7.20) | |
| Smoking status (vs never smoked) | | | | |
| Current smoker | 1.37 (0.70 to 2.71) | 0.563† | – | – |
| Ex-smoker | 0.95 (0.70 to 1.28) | | – | |

N=1008 in univariable models unless otherwise stated; N=1004 for final multivariable logistic regression model.
*Adjusted for age, sex, deprivation level, number of comorbidities, estimated glomerular filtration rate (eGFR) at 5-year follow-up, albumin–creatinine ratio, educational attainment and body mass index (BMI).
†P value for trend.
GCSE, General Certificate of Secondary Education; KDIGO, Kidney Disease Improving Global Outcomes; A level, advanced level; NVQ, National Vocational Qualifications; uACR, urine albumin–creatinine ratio.

and to discuss these with patients and develop individualised management plans. Among other things, they suggest that clinicians assess for frailty (using measures such as gait speed or self-reported health status) and remain particularly vigilant for issues with mental health or chronic pain.[31] Our data provide evidence that a similar approach is warranted in people with mild-to-moderate CKD. When managing these patients, clinicians should consider comorbidities and discuss how treatments will fit in with those for other comorbidities, particularly mental health and chronic pain-related conditions and those affecting mobility. Obesity, as a potentially modifiable factor with an independent association with both functional impairment and HRQoL in our study, is also an important consideration. This could include devising management plans jointly with other healthcare professionals, streamlining health services to cater for people with multiple conditions, and signposting to appropriate services. Both general practitioners and nephrologists should be well placed to manage patients holistically, but patients would benefit from mental health, comorbidity and pain issues being considered in clinical guidelines and outcome measures. It is also important to consider the capacity of people to self-manage, and the extent of patient activation, if mental health issues or multiple comorbidities are present.

## CONCLUSION
We have observed that people with mild-to-moderate CKD commonly have multiple comorbidities and many report some HRQoL problems and functional impairment. Taken together with previous observations that the risk of CKD progression is low in this context, these data suggest that mild-to-moderate CKD in older people could be more important as a marker of increased comorbidity

than as a risk factor for ESKD. Clinicians should therefore carefully consider comorbidities and discuss how treatments will fit in with those for other comorbidities, particularly mental health, chronic pain-related conditions and those affecting FS.

**Acknowledgements** The authors acknowledge the contributions of the participants, their families and the participating general practice surgeries without whom this work would not have been possible. The authors would like to thank Shihua Zu who provided advice on using EQ-5D-5L.

**Contributors** SF, JB, PR, JEM and HMY designed and undertook the analyses. AS and MT obtained the cohort 5-year follow-up data. MT, NM, RF and CWM established the RRID cohort, provided data and contributed development of the project and critically reviewed the manuscript. All authors critically reviewed the paper; were involved in the drafting and approval of the final manuscript; and act as guarantors. All authors take responsibility for the data and research governance.

**Funding** The RRID study was funded by a Research Project Grant (R302/0713) from the Dunhill Medical Trust (http://www.dunhillmedical.org.uk). Previous study funding includes a joint British Renal Society (http://www.britishrenal.org) and Kidney Research UK (http://www.kidneyresearchuk.org) fellowship (BRS3/2008, to NJM), and an unrestricted educational grant (EPWE124712-G) from Roche Products Ltd. (https://www.roche.co.uk).

**Disclaimer** The funders had no role in study design, data collection and analysis, decision to publish, or preparation of the manuscript.

**Patient consent for publication** Not required.

**Ethics approval** The RRID study was approved by the Nottingham Research Ethics Committee 1 and was included on the National Institute for Health Research (NIHR) Clinical Research Portfolio (NIHR Study ID: 6632). All participants provided written, informed consent. The RRID study complies with the Declaration of Helsinki and the principles of Good Clinical Practice.

**Provenance and peer review** Not commissioned; externally peer reviewed.

**Data availability statement** Data are available upon reasonable request. Anonymised data can be made available to researchers who meet the conditions of the ethics approval and research governance policy that applies to this study. Researchers may apply for data access by contacting Dr. Teresa Grieve, Research and Development Manager, University Hospitals of Derby and Burton NHS Foundation Trust (teresa.grieve@nhs.net).

**Author note** We affirm that the manuscript is an honest, accurate, and transparent account of the study being reported. No important aspects of the study have been omitted and any discrepancies from the study as originally planned have been explained.

**ORCID iDs**
Simon DS Fraser http://orcid.org/0000-0002-4172-4406
Maarten W Taal http://orcid.org/0000-0002-9065-212X

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
