## [Reviewer comments · BMJ Open]

ARTICLE DETAILS

TITLE (PROVISIONAL)	Health related quality of life, functional impairment and comorbidity in people with mild to moderate chronic kidney disease: a cross sectional study
AUTHORS	Fraser, Simon; Barker, Jenny; Roderick, Paul; Yuen, Ho Ming; Shardlow, Adam; Morris, James; McIntyre, Natasha; Fluck, Richard; McIntyre, CW; Taal, Maarten

VERSION 1 – REVIEW

REVIEWER	Marjorie Foo Singapore General Hospital Singapore
REVIEW RETURNED	04-Jun-2020

GENERAL COMMENTS	The paper is well written and findings are consistent with other papers published in this area of QOL in CKD. The assessment of QOL using ED 5D is a safe tool . The analysis is fine and conclusion appropriate . The limitation of the study has been elaborated in the paper by the authors. Of note, the population of CKD with DM appears low, for UK population , the sample is also very homogenous with Caucasian dominance. Optional to add in comments to the paper . Overall , well written
---

REVIEWER	Mariangela Leal Cherchiglia Universidade Federal de Minas Gerais, Brazil
REVIEW RETURNED	05-Jun-2020

GENERAL COMMENTS	The article is well written, the text is easy to understand and presents the results of health-related quality of life and functional status in a large cohort of people with mild to moderate CKD recruited from primary care in the UK. I would like to congratulate the authors for the extensive and important data collection that was carried out to develop and present this study. However, trying to increase the study quality, I have a few issues to recommend: In the Results, Table 1, it is observed that in some variables (for example, age and education), the percentages add up to less or more than 100%. Please check the percentages and correct them. In the Results, Table 2 shows a comparison between groups, but does not show any results from statistical tests. The same happens in Table S1. Therefore, I would like to know why the authors only made a descriptive analysis in this comparison. (Table 2 and Table
---

	S1). In the second paragraph of the discussion (lines 21-50), the authors present and discuss the study's strengths and limitations. I suggest that this topic be placed at the end of the discussion
--	---

VERSION 1 – AUTHOR RESPONSE

Reviewer: 1

Reviewer Name: Marjorie Foo

Institution and Country: Singapore General Hospital, Singapore

Please state any competing interests or state 'None declared': None declared

The paper is well written and findings are consistent with other papers published in this area of QOL in CKD. The assessment of QOL using ED 5D is a safe tool . The analysis is fine and conclusion appropriate .

Authors' response: Thank you for your positive assessment of our paper

The limitation of the study has been elaborated in the paper by the authors.

Of note, the population of CKD with DM appears low, for UK population, the sample is also very homogenous with Caucasian dominance.

Optional to add in comments to the paper.

Authors' response: Thank you for your comment on this – we have added a sentence to the discussion to make this point:

'The prevalence of diabetes in this population of people with CKD stage 3 was 14.2%. This is lower than the prevalence of 20.1% noted in analyses of CKD prevalence in the Health Survey for England.[2] It is possible that this relates to this study population comprising survivors at five years in a cohort study and those with diabetes may have been more likely to die prior to these analyses than those without. The study population was also predominantly white and those ethnic groups with greater diabetes prevalence were therefore under-represented.'

Overall, well written

Authors' response: Thank you

Reviewer: 2

Reviewer Name: Mariangela Leal Cherchiglia

Institution and Country: Universidade Federal de Minas Gerais, Brazil

Please state any competing interests or state 'None declared': None declared

The article is well written, the text is easy to understand and presents the results of health-related quality of life and functional status in a large cohort of people with mild to moderate CKD recruited from primary care in the UK. I would like to congratulate the authors for the extensive and important data collection that was carried out to develop and present this study.

Authors' response: Thank you for your positive assessment of our paper

However, trying to increase the study quality, I have a few issues to recommend:

In the Results, Table 1, it is observed that in some variables (for example, age and education), the

percentages add up to less or more than 100%. Please check the percentages and correct them.

Authors' response: Thank you for noticing this potential discrepancy. We have checked the figures in Table 1 and, where the total is more or less than 100%, this was due to rounding the values. For example 30.45%+69.55% would be rounded to 30.5%+69.6% which would be more than 100%. We could force the numbers to add up to exactly 100% but strictly speaking we believe that this would be inaccurate. We have therefore added the following sentence to the footnote of the table: 'Where variable category percentages sum to less than or more than 100%, this is due to rounding.'

In the Results, Table 2 shows a comparison between groups, but does not show any results from statistical tests. The same happens in Table S1. Therefore, I would like to know why the authors only made a descriptive analysis in this comparison. (Table 2 and Table S1).

Authors' response: Thank you for raising this issue. The lack of statistical comparisons in Tables 2 and S1 were something we considered carefully and made a deliberate decision to only present the descriptive analyses. In the case of Table 1 it was felt that this was a descriptive comparison with Health Survey for England data where the HSE sample were not data collected by us and we felt that to give a statistical comparison might artificially suggest that we had collected the data ourselves. Furthermore, we felt that the populations (RRID and HSE), while similar enough to merit a descriptive comparison, were not identical in several important ways (e.g. the characteristics of the samples, the way the data were collected, the context of each study etc) and felt that making statistical comparison was therefore somewhat artificial and arguably invalid. In Table S2 we felt that it was more valuable for the reader to come to their own decision about the degree to which our sample population differed from the population with incomplete follow up data.

We would be prepared to amend either or both of these decisions if statistical comparison is favoured by the reviewer and/or the editor in this situation.

In the second paragraph of the discussion (lines 21-50), the authors present and discuss the study's strengths and limitations. I suggest that this topic be placed at the end of the discussion

Authors' response: We understand that this is a reasonable request, and common practice in many journals. However, we placed the strengths and limitations here based on the BMJ Open guidelines for authors (here: <https://bmjopen.bmj.com/pages/authors/#research>) which suggests placing the strengths and limitations after the statement of principal findings:

'We also recommend, but do not insist, that the discussion section is no longer than five paragraphs and follows this overall structure (you do not need to use these as subheadings): a statement of the principal findings; strengths and weaknesses of the study; strengths and weaknesses in relation to other studies, discussing important differences in results; the meaning of the study: possible explanations and implications for clinicians and policymakers; and unanswered questions and future research.'

Again – we would be quite happy to move the strengths and limitations section to later in discussion if that is the editorial preference but for the moment we suggest leaving it in its current position based on these author guidelines.

VERSION 2 – REVIEW

REVIEWER	Mariangela Leal Cherchiglia Universidade Federal de Minas Gerais, Brazil
REVIEW RETURNED	06-Jul-2020

GENERAL COMMENTS	The authors' adaptations and responses were evaluated positively, improving the quality of the manuscript. The authors answered each question raised by the editor and academic reviewers.
--